# The Adoption of Artificial Intelligence in Serbian Hospitality: A Potential Path to Sustainable Practice

Tamara Gajić [1,2,*], Dragan Vukolić [2,3], Jovan Bugarčić [2], Filip Đoković [4], Ana Spasojević [5], Snežana Knežević [6], Jelena Đorđević Boljanović [4], Slobodan Glišić [7], Stefana Matović [1] and Lóránt Dénes Dávid [8,9]

1. Geographical Institute "Jovan Cvijić" SASA, 11000 Belgrade, Serbia; s.babovic@gi.sanu.ac.rs
2. Faculty of Hotel Management and Tourism, University of Kragujevac, 36210 Vrnjačka Banja, Serbia; vukolicd@yahoo.com (D.V.); bugarcicjovan@gmail.com (J.B.)
3. Faculty of Tourism and Hospitality, University of Business Studies, 78000 Banja Luka, Bosnia and Herzegovina
4. Faculty of Organizational Studies—EDUKA, 11000 Belgrade, Serbia; fdjokovic@vos.edu.rs (F.Đ.); jelenaboljanovic@vos.edu.rs (J.Đ.B.)
5. Department for International Cooperation, University of Kragujevac, 34000 Kragujevac, Serbia; ana.spasojevic1985@gmail.com
6. Department of Medical Studies, Academy of Applied Technical Studies Belgrade, 11000 Belgrade, Serbia; lesta59@yahoo.com
7. Academy of Professional Studies South Serbia, 16000 Leskovac, Serbia; glisictfl@gmail.com
8. Department of Tourism and Hospitality, Faculty of Economics and Business, John von Neumann University, HU-6000 Kecskemét, Hungary; david.lorant.denes@uni-neumann.hu
9. Department of Sustainable Tourism, Institute of Rural Development and Sustainable Economy, Hungarian University of Agriculture and Life Sciences (MATE), HU-2100 Gödöllő, Hungary
* Correspondence: tamara.gajic.1977@gmail.com

**Abstract:** This study investigates the perceptions of employees in the hotel industry of the Republic of Serbia regarding the acceptance and importance of artificial intelligence (AI). Through a modified UTAUT model and the application of structural equation analysis (SEM), we investigated the key factors shaping AI acceptance. Research results show that behavioral intention and habit show a significant positive impact on AI usage behavior, while facilitating conditions have a limited but measurable impact on behavioral intention. Other factors, including social influence, hedonic motivation, performance expectancy, and effort expectancy, have minimal influence on the examined variables. The analysis reveals the crucial mediating role of behavioral intention, effectively bridging the gap between various predictors and AI usage behavior, thereby highlighting its significance in the broader context of technology adoption in the hotel industry. The primary goal of the study, which closes significant research gaps, as well as the manner in which it uses a specific model and statistical analysis to accomplish this goal, shows how innovative the work is. This method not only broadens the field's understanding but also offers valuable insights for shaping sustainable development practices in the hospitality sector in the Republic of Serbia.

**Keywords:** artificial intelligence; employee attitudes; sustainability; hotel industry; Republic of Serbia

## 1. Introduction

The hotel industry globally is at the forefront of adopting innovative technologies to improve guest experiences and operational efficiency. Among these technologies, artificial intelligence (AI) has emerged as a transformative force, expanding service offerings and redefining operational strategies. The incorporation of AI into the hospitality sector presents numerous opportunities and challenges, particularly in the developing Serbian hotel industry, which is navigating its path to digital transformation amidst unique economic and cultural contexts. This backdrop provides a compelling setting to explore how AI

technologies are being adopted and their impacts on hotel operations and guest experiences within Serbia. Although the global hospitality sector has made significant progress in the application of AI, there continues to be an obvious gap in our understanding of how hotel employees in Serbia experience AI and its implications for sustainable business [1–5]. This gap is critical because employee attitudes significantly dictate the success and sustainability of AI technology's integration. Acknowledging this, our study aims to fill this gap by providing an in-depth analysis of the attitudes of employees in the hotel industry in Serbia towards the adoption of AI, with a particular focus on its potential sustainability implications in the hotel sector in the future. In this research, the main target variables are the AI usage behavior, which describes the actual behavior of the user in connection with the use of AI for the purpose of sustainability, and the behavioral intention, which describes the intention or will of the user toward the future use of AI with the aim of potential sustainability. To address the identified gap, our study aims to explore the following key questions:

Q1. What are the perceptions of employees in the Serbian hotel industry regarding the acceptance and use of AI?

Q2. How do these perceptions influence their actual usage behaviors towards AI technologies?

Q3. What role do the facilitating conditions, effort expectancy, performance expectancy, social influence, and hedonic motivation play in shaping these perceptions and behaviors?

Q4. To what extent can the adoption of AI in the Serbian hotel industry contribute to its sustainability goals?

The motivation for this research is twofold. First, we seek to uncover the under-researched dimensions that influence the adoption of AI in the Serbian hotel industry, offering insights that go beyond the current literature. This research is crucial, bearing in mind the unique socio-economic and technological space of Serbia, which can affect the perception and behavior of employees differently compared to other regions. Second, we aim to establish a thorough understanding of how AI can drive sustainable practices in the hospitality industry. It is important to clarify that while our study explores the potential for sustainability through AI, it does not claim to establish definitive sustainability outcomes from AI integration. Instead, our research sets the stage for future research on how artificial intelligence can improve sustainable practices in the hospitality sector.

The significance of our study is manifold. Theoretically, it enters the relatively unknown territory of the adoption of AI in the Serbian hotel industry, thus enriching the academic approach with new insights. Practically, it provides valuable information to hotel industry managers, policymakers, and scholars, guiding the development of effective strategies for the application of AI, employee training programs, and organizational changes adapted to a Serbian context. Our study stands out in that it focuses on the Serbian hospitality sector, comprehensively examines employee attitudes towards AI, and explores the potential benefits of AI in terms of sustainability. Through rigorous methodological approaches, including detailed statistical analyses and structural equation modeling, we ensure the reliability and validity of our findings, contributing to a solid framework for future research on AI adoption in hotel organizations.

This study not only addresses a significant gap in the literature but also focuses on the unique dynamics of the adoption of AI in the Serbian hospitality industry. As AI continues to evolve in the hospitality sector, our research offers fundamental insights that will inform future academic research and strategic decisions, ensuring that the Serbian hospitality industry remains competitive and sustainable in the global marketplace.

## 2. Theoretical Background and Hypothesis

According to Martínez et al. [6], the hospitality sector plays a crucial role in the economic prosperity of numerous states. The provision of food, beverages, and lodging is the main focus of hospitality services, which can be provided in both commercial and

non-commercial settings [7]. Because the hospitality industry is based on providing human services, it is heavily reliant on representation and customer reviews [8]. Artificial intelligence (AI) is the process by which computer systems mimic human intelligence functions [9]. According to Roussel and Norvig [10], AI has evolved into a powerful force that has a significant impact on a variety of business aspects across all industries. The hospitality industry, which includes hotels and tourism, food and beverage, and meeting and event businesses, makes extensive use of AI and robotic technologies [11]. In the food and beverage industry, sale controls can be handled by the artificial intelligence used in point-of-sale systems [12]. Automation and AI services not only decrease human error but also forecast future business success [13]. Automation is actually fueled by machine learning algorithms, which boost operational efficiency in the hotel industry significantly [14]. This is a strategic step towards optimizing and sustainable processes, but it also presents a challenge for cost reduction and a notable improvement in the decision-making process [15]. According to Davenport and Harris [16], AI provides data-driven insights, allowing businesses to make more informed strategic decisions. This capability is further emphasized by Provost and Fawcett [17], who highlight the crucial role of predictive analytics in enabling executives to predict market trends, optimize supply chains, and make swift decisions by identifying patterns and correlations in large datasets. Through real-time data analysis, AI is being applied in the hotel industry to help businesses gain a competitive edge in dynamic markets and quickly adapt to changing conditions [18]. AI integration aims to achieve three major objectives: customization, anticipating client demands, and increasing pleasure through unique experiences [19]. It also aims to optimize operational procedures and revenue management and adapt service prices to reflect changes in the market [20]. The automation of routine tasks, such as reservation and inventory management, contributes to more efficient hotel operations [21]. A key motivation for adopting AI in the hospitality industry stems from the desire to improve the guest experience [22]. However, implementing AI in the hotel industry presents challenges, particularly in managing extensive personal guest data, leading to concerns about data privacy [23–27]. Hotels struggle to maintain compliance with rules and protect the privacy of personal information [28,29]. Additionally, the integration of AI requires its successful alignment with existing technological systems, demanding organizational and technical expertise [30]. Despite these challenges, AI enhances security measures by leveraging features such as facial recognition and real-time security situation monitoring, thereby expanding its advantages [31–33].

In the modern hotel industry, employee attitudes regarding AI play a crucial role in shaping the use of AI in the workplace, directly impacting sustainability within the sector [34]. The divergent attitudes and expectations of employees impact the adoption process of AI, thereby exerting an influence on the social, environmental, and economic dimensions of sustainability [35–37]. The automation of work raises concerns about potential job losses and the erosion of specialized skills, yet it is also recognized for its potential to enhance productivity and stimulate creativity, contributing to economic sustainability and resource efficiency [38,39]. Those with a positive outlook see AI as a tool that not only enhances working conditions but also opens up new opportunities for career advancement, thereby supporting social sustainability by fostering a skilled and adaptable workforce [40]. Importantly, attitudes towards AI vary according to the industry and job function, with employees in sectors that are more integrated with AI, such as information technology, being more open to its benefits, including its contributions to sustainability by optimizing operations and reducing waste [41–43]. Through strategies such as open communication, employee training, and policies that encourage human–AI collaboration, organizations can significantly influence attitudes towards AI, overcoming resistance and building trust [37]. This comprehensive approach ensures the balanced integration of AI, highlighting its positive effects in the workplace while also emphasizing its role in advancing sustainability in the hotel industry [44–47].

### 2.1. The Role of the Unified Theory of Acceptance and Use of Technology (UTAUT) in Developing AI Integration Models in Hospitality

A variety of theoretical vantage points, including the cognitive, affective, motivational, and behavioral intentions and reactions of individuals, have contributed to the evolution of the acceptance of contemporary technologies in business. The most commonly used theory in the study of user behavior towards the acceptance of artificial intelligence is the Unified Theory of Acceptance and Use of Technology (UTAUT) [48]. This theory represents a unique synthesis of various theories of user behavior's study, including the Theory of Reasoned Action (TRA), the Theory of Planned Behavior (TPB), the Technology Acceptance Model (TAM), the combined form of TAM and TPB (C-TAM-TPB), the Model of PC Utilization (MPCU), Innovation Diffusion Theory (IDT), the Motivational Model (MM), and Social Cognitive Theory (SCT) [49]. Developed by Venkatesh et al. [50], UTAUT provides a holistic view by incorporating elements such as social impact and performance expectancy.

The development of AI integration models in the hotel sector is greatly aided by the Unified Theory of Acceptance and Use of Technology. In order to forecast and explain behavior towards AI adoption, this framework emphasizes the significance of performance expectancy, effort expectancy, social influence, and facilitating settings. These factors are essential for the long-term integration of technology in hotels [51]. By applying UTAUT, hospitality businesses can gauge the likelihood of successful AI implementation, addressing workforce concerns and expectations, thereby facilitating a smoother transition to technologically advanced, sustainable operations [52,53]. This model is key in directing strategic planning and training initiatives for AI deployment, ensuring the technology is not only effectively utilized but also that the workforce is fully prepared and supportive, aligning with economic, social, and environmental sustainability goals [54].

Some authors emphasize that each theory has its limitations and does not complement each other completely [55]. There are differences in terminology between them, but they are essentially directed towards the same concepts. Also, given the complexity of studying behavior and the limitations of researchers, there is no universal theory that encompasses all behavioral factors [56–58].

### 2.2. Behavioral Intention and AI Usage Behavior in the Hotel Industry

The application of AI to hotel business is a game-changing development that promises increased productivity, better customer satisfaction, creative operational procedures, and sustainable business in the field [59]. As organizations navigate this technological evolution, it is imperative that they understand the multiple factors shaping AI adoption among hotel industry employees in Serbia. Behavioral intent is emerging as a key factor influencing AI adoption [60]. The recognition of their perceived ease of use and benefits, according to Davis [61], significantly influences users' intentions to adopt new technologies. According to his claim, in the hotel industry, the attitudes and intentions of employees play a key role in shaping their willingness to adopt artificial intelligence technologies. Knowing the drivers of these behavioral intentions is essential to formulating strategies that are in line with employee expectations and preferences [62]. Positive factors can explain why people engage in certain behaviors, but they cannot predict the reasons behind their resistance to those behaviors [63]. Therefore, it is important to identify the negative factors that influence people to refrain from specific behaviors. Theoretically, these subjective elements may differ depending on how clients view their values and beliefs [64–66]. There are a number of factors that can influence one's ability to positively influence employee attitudes, and their justifications can offer insights into contextual or situational decision-making [67]. In this context, the Unified Theory of Acceptance and Use of Technology (UTAUT) further enriches the understanding of intentions and behaviors related to the adoption of artificial intelligence in hotels. Thus, we proposed the next hypothesis.

**H1.** *Behavioral intention positively mediates the relationship between performance expectancy, effort expectancy, hedonic motivation, social influence, facilitating conditions, habits, and AI usage behaviors.*

### 2.3. Facilitating Conditions and the Acceptance of AI in the Hotel Industry

Creating facilitating conditions is key to fostering positive attitudes towards the adoption of artificial intelligence (AI). Venkatesh et al. [54] elaborate on the concept of facilitating conditions from the Technology Acceptance Model (TAM), emphasizing the influence of organizational support and resources on user intentions and behaviors. Critical determinants, such as adequate resources, support systems, and infrastructure, are encompassed within these facilitating conditions. This is in line with the understanding that when organizations provide the necessary conditions, employees are more likely to express positive intentions towards adopting AI. According to Davenport [16], facilitating conditions encompass the organizational and technical support that can ease in the integration of AI technologies. The availability of favorable facilitating conditions significantly affects the behavioral intentions of employees towards artificial intelligence and the creation of a positive attitude that AI can significantly contribute to sustainability in a hotel business. In other words, when organizations provide adequate resources, training, and support for AI adoption, employees are more likely to develop positive attitudes and intentions towards incorporating AI technologies into their work processes, as well as creating conditions for potential future sustainability. Venkatesh et al. [55] emphasize that facilitating conditions not only encourages favorable attitudes but also have a significant positive effect on AI usage behavior. When employees understand that there are necessary conditions for a sustainable business, they are more likely to actively engage and use AI technologies in their daily tasks. This continuity in considering the factors that shape employees' attitudes and intentions towards artificial intelligence illustrates the importance of the UTAUT model (Unified Theory of Technology Acceptance and Use). Based on the conclusion that the availability of favorable enabling conditions is significant, we made an assumption.

**H2.** *Facilitating conditions have a significant impact on employees' behavioral intention towards AI in the Serbian hotel industry.*

### 2.4. Hedonic Motivation

Hedonic motivation is also introduced by Momani et al. [68] as a key element that influences the intentions and behaviors of employees. Employee intentions are influenced by the satisfaction, fun, and experiential aspects of interacting with AI technologies, which also create a positive and engaging experience when adopting AI [69]. In the context of the hotel sector, people are more inclined to accept AI technologies when they have enjoyable and positive experiences using them, knowing that they support the development of the hotel industry in a sustainable way [70]. Artificial intelligence-powered personalized recommendations, interactive services, and smart room features can all lead to a more enjoyable and rewarding hotel stay, which in turn encourages guest loyalty and further suggests sustainability in the business setting [71–74]. These are some instances of hedonic motivation in action. Employees are more hedonistically motivated to adopt artificial intelligence when they believe that these technologies will improve their work experience or improve customer satisfaction, thereby creating a continuity of business with loyal visitors [75–77]. More precisely, this favorable correlation with pleasant experiences may increase the willingness to accept and implement AI technologies in different contexts of sustainability in hotel operations. In light of the aforementioned data, and based on the above-mentioned findings, we hypothesize that:

**H3.** *Hedonic motivation has a significant positive effect on the behavioral intention towards AI in the Serbian hotel industry.*

### 2.5. Performance Expectancy

Focusing on the expected performance of artificial intelligence (AI) in the hotel industry, Davies [61] highlights the importance of employee perceptions of the technology's usefulness in creating a long-term or sustainable business. Regarding artificial intelligence, how stakeholders and employees view the technology's potential benefits is a major factor in determining how likely they are to adopt it. Employees and other stakeholders are more ready to embrace AI technology if they believe it may greatly increase operational efficiency, lead to better outcomes, foster long-term business relationships, and raise customer satisfaction [78]. Employee expectations regarding positive results and performances, if met, can form positive intentions to adopt and use AI technologies in hotel processes, which further contribute to sustainable business. Gaining insight into the expected performance of AI technologies is essential to advance the adoption of these technologies into the hotel industry [79]. Positive perceptions can help overcome resistance to change and create a positive environment for AI integration, which can ultimately lead its to successful adoption and deliver significant benefits [80]. The following hypothesis was established:

**H4.** *Performance expectancy has a significant positive effect on behavioral intention.*

### 2.6. Effort Expectancy

The expected effort in the use of AI in the hotel industry is a key component of the successful adoption of these technologies. Venkatesh et al. [55] emphasize the importance of this factor, showing the influence of users' perception of ease of use on their intentions and behaviors. Designing easy-to-use AI solutions is an important aspect, bearing in mind the need to seriously consider user needs and expectations. Expected effort plays a key role in the formation of employee intentions in the hotel industry, which highlights the need for AI solutions that are easy to use, if employees are aware of the importance of AI for sustainable business [81]. The user's awareness of and ability to perceive the technology as adapted to the sustainability system of the hotel business and as being easy to use significantly affects their willingness to adopt and use AI technologies [82]. The original definition of expected effort duration was proposed by Davies [61] and evaluated the usability of the technology. When it comes to artificial intelligence, the easier it is for employees and stakeholders to integrate these technologies into their work, the more likely they are to be seen as adaptable and useful for the future, requiring little effort or complexity to use [83–85]. Years of research by Ankara and Walden [86] provide additional proof that people's perceptions of artificial intelligence systems' simplicity and ease of use have a substantial impact on their acceptability. If employees think that using artificial intelligence will be simple and require little effort, and, on the other hand, contribute to the sustainability of the business, their willingness to incorporate it into their tasks will positively affect their perception [87].

**H5.** *Effort expectancy has a significant positive effect on behavioral intention.*

### 2.7. The Influence of Habits on the Acceptance of AI by Hotel Employees

Ingrained habits play a significant role in the adoption process of AI by hospitality employees. Cain et al. [88] point out that habitual behaviors not only influence intentions but also the consistent and repetitive use of AI technologies. Leveraging existing habits becomes key to successful integration strategies. Respecting the role of habits, which are constantly present in employees' daily tasks, it is important to design AI integration so that it does not disrupt existing routines [89]. If artificial intelligence is harmoniously integrated into workflows with evidence of a positive effect on sustainability being at work, and with it being adapted to existing habits, the chances of successful acceptance by employees increase significantly [90]. Studies indicate that the development of positive habits related to the use of artificial intelligence can be achieved through targeted training and familiarization programs [91–93]. Supporting employees in developing habitual patterns of interaction with AI systems through the regular use of the tool, for example, for data analysis, customer

interaction, or room management, can increase their comfort level and enable its successful adoption [94]. Also, emphasis is placed on the organizational culture and its role in shaping habits related to technological innovation. A culture that encourages experimentation, learning, and the integration of artificial intelligence into daily routines contributes to the creation of positive habits among employees and the successful acceptance of these technologies [95]. Emphasizing the role of ingrained habits in the acceptance of AI, an assumption is made.

**H6.** *Habit has a significant positive effect on behavioral intention.*

*2.8. Social Influence*

Interpersonal factors, including relationships with colleagues and superiors, play a crucial role in the acceptance process of AI by employees in the hotel industry [96]. Introducing social aspects, such as subjective norms and social influences, is a critical element for crafting strategies that support collaborative dynamics in the hotel industry [97]. The attitudes of employees are significantly influenced by the social consequences of the adoption of AI, which implies an awareness of the social determinants of business. Concerns about job displacement, changes in team dynamics, or changes in job responsibilities can profoundly affect a technology's acceptance [98]. Therefore, proactive measures, such as open communication, developing an awareness of environmental and social sustainability, training programs and involving employees in decision-making processes, are essential for increasing social acceptance [99]. Additionally, numerous social risks, including privacy and the ethical use of data, underscore the need for clear rules and standards. Responsibility in handling data and the ethical use of AI technologies are key factors influencing employee acceptance [100]. Thus, improving the social environment within the company and fostering positive peer relationships can facilitate the acceptance process [101]. Ongoing feedback and channels for expressing employees' opinions about AI technologies are crucial aspects. This open and inclusive strategy empowers employees to actively participate in defining the organizational AI landscape and effectively addressing social challenges [102]. Based on these ideas, which have social implications, we present the following hypothesis (Figure 1).

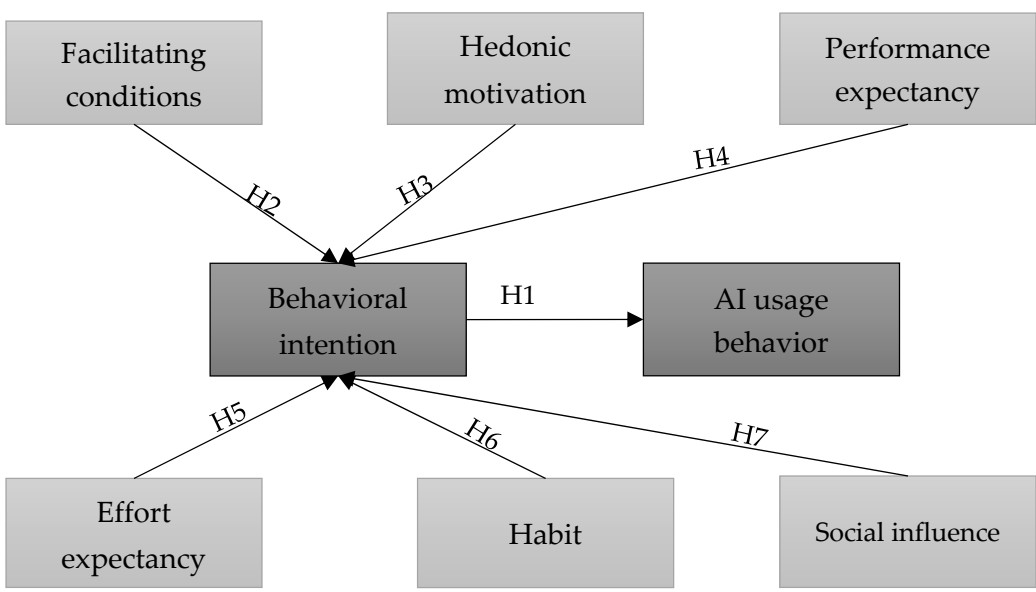

**Figure 1.** Research Model Framework.

**H7.** *Social influence has a significant positive effect on behavioral intention.*

## 3. Methodology

### 3.1. Research and Questionnaire Design

The methodology included several key components, including construct measurement, data collection, and statistical analysis. The survey instruments were designed based on the established model of the Unified Theory of Acceptance and Use of Technology (UTAUT) used by many authors in their research [28,47,51,53,57]. The questions were taken and modified specifically from the research of Ali et al. [103] and Vinesa et al. [104]. Identified constructs included behavioral intention (4 items), facilitating conditions (4 items), hedonic motivation (4 items), performance expectancy (4 items), AI usage behavior (3 items), effort expectancy (4 items), habit (4 items), and social influence (4 items). These constructs were chosen based on their relevance to understanding the factors influencing AI adoption in hotel settings. Constructs were operationalized using validated measurement items. Items were carefully selected to capture the essence of each construct, ensuring content validity. It must be noted that AI usage behavior describes the actual behavior of users in connection with the use of artificial intelligence in the direction of sustainability, and behavioral intention describes the user's intention or willingness to use AI in the future with the idea of potential sustainability.

### 3.2. Sample and Data Collection Procedure

Stratified random sampling was the method of sampling that was used in this investigation. Based on their employment duties, the population of interest, which was made up of workers in Serbian hotel businesses, was separated into strata. Participants were chosen at random from each stratum to guarantee representation from a variety of industry roles. The chosen participants were subsequently surveyed in-person to gather data, and a total of 479 complete responses were obtained. The study was conducted at seven five-star hotels in Serbia, in the two main cities of the Republic of Serbia, Novi Sad and Belgrade, between May and August of 2023. Measurement items were created for every construct in an organized survey instrument. Respondents gave their perceptions and attitudes on a five-point Likert scale (1—strongly disagree, 2—disagree, 3—neither agree nor disagree, 4—agree, 5—strongly agree), reflecting their intensity of agreement or disagreement. Men dominate among the respondents, and the largest age group is that of people between 20 and 35 years old, with university education being the most represented, and the majority of respondents having incomes between 500 and 1000 euros (Table 1).

**Table 1.** Socio-demographic characteristics of the respondents.

| Gender | | Age | | Education | | Monthly Income (in Euros) | |
|---|---|---|---|---|---|---|---|
| Men | Women | 20–35 | 38.2% | High School | 29.8% | <500 | 2.7% |
| 54.1% | 45.9% | 36–60 | 36.7% | Faculty | 60.2% | 500–1000 | 62.3% |
| | | >61 | 25.1% | PhD, MSc | 10% | >1000 | 35% |

### 3.3. Data Analysis

To describe the sample and comprehend the central tendency and variety of responses, descriptive statistics such as means, standard deviations, and frequency distributions were utilized. After dividing the items into factors using a factor analysis, eight factors were discovered. To verify the robustness of the factor analysis and the presence of these eight factors, Horn's parallel method was used [105]. The Kaiser–Meyer–Olkin (KMO) [106] measure indicates a high value of 0.830, suggesting that the sampling adequacy for the factor analysis is good. This implies that the collected data is suitable for extracting meaningful factors. Bartlett's test of sphericity [107], with an approximate Chi-Square value of 7309.858 and 105 degrees of freedom, is statistically significant ($p = 0.000$). This indicates that correlations between variables are sufficiently different from zero, justifying the use of factor analysis. Among the identified factors, performance expectancy explained the largest

percentage of variance, accounting for 24.51%. On the other hand, AI usage behavior explained the least percentage of variance, with a value of 3.53%. The total percentage of explained variance across all factors was 66.31%.

Structural equation modeling (SEM) was utilized to assess the relationships among the identified constructs [108]. The model aimed to explore the predictive power of various factors on behavioral intention and AI usage behavior. A path analysis was conducted to estimate the direct and total effects of each factor [109]. Cronbach's alpha, rho_A, composite reliability, and the average variance extracted (AVE) were calculated to evaluate the reliability and validity of the measurement model [108]. Discriminant validity was assessed using the Heterotrait–Monotrait Ratio (HTMT) and the Fornell–Larcker criterion [110]. Variance inflation factor (VIF) values were examined to identify potential multicollinearity issues among the predictor variables. Fit indices, including SRMR, d_ULS, d_G, Chi-Square and NFI were employed to assess the overall goodness of fit of the SEM model [111]. A bootstrapping analysis was conducted to verify the identified relationships. The software SmartPLS 3 was used to carry out structural equation modeling (SEM).

The answers to questions about attitudes and perceptions regarding the use of artificial intelligence (AI) in the context of sustainable business practices in the hotel industry are shown in Table 2. A single AI usage item is represented by each row, and the columns provide the mean (m) and standard deviation (sd) of replies, the factor loading that indicates the item's relationship to the underlying construct, and the Cronbach's alpha ($\alpha$) value for internal consistency reliability. The good internal consistency of the measures is demonstrated by the results, which are backed by high values of Cronbach's alpha coefficient ($\alpha$) for every construct. Furthermore, a robust measurement model is shown by the fact that the detected factors account for a considerable percentage of response variance, ranging from 81.9% to 93.1%.

**Table 2.** The measurement items, descriptive statistics, and reliability analysis of the constructs.

| Items | m | sd | $\alpha$ | Factor Loading |
|---|---|---|---|---|
| AI is beneficial to sustainable hotel business | 2.03 | 1.168 | 0.852 | 0.659 |
| AI helps to complete the task faster | 2.97 | 1.437 | 0.843 | 0.787 |
| AI brings convenience to my work | 2.25 | 1.357 | 0.846 | 0.701 |
| AI can improve the sustainability of service quality | 3.09 | 1.382 | 0.825 | 0.744 |
| We are ready to use AI because it is easy to understand | 3.09 | 1.382 | 0.818 | 0.623 |
| Using the AI interface is less complex | 2.14 | 1.325 | 0.847 | 0.841 |
| The AI is intuitive and efficient to use | 2.12 | 1.325 | 0.868 | 0.744 |
| AI makes it easier for me to become an expert/skilled | 2.13 | 1.292 | 0.856 | 0.559 |
| People around me think that artificial intelligence should be used for business sustainability | 2.56 | 1.440 | 0.832 | 0.719 |
| Family and friends have an important role to play in the use of artificial intelligence | 2.01 | 1.291 | 0.876 | 0.658 |
| The use of artificial intelligence seems prestigious/admirable during travel | 4.35 | 2.205 | 0.841 | 0.743 |
| I will discuss the feeling of using artificial intelligence when traveling with my family | 3.86 | 2.068 | 0.836 | 0.783 |
| We can afford digital devices to use artificial intelligence | 3.52 | 2.089 | 0.839 | 0.827 |
| People around me think that artificial intelligence should be used for business sustainability | 2.56 | 1.440 | 0.832 | 0.719 |
| I have the necessary resources to use AI | 3.07 | 1.973 | 0.811 | 0.709 |
| AI is compatible with the technology devices I use | 3.27 | 1.994 | 0.829 | 0.699 |
| I can get help from others when I have difficulty using AI | 3.81 | 2.144 | 0.840 | 0.646 |
| Using artificial intelligence is fun for me because I contribute to sustainable business and quality | 3.03 | 1.950 | 0.891 | 0.759 |
| I like the AI application | 3.68 | 2.070 | 0.844 | 0.694 |
| AI application is kind of fun for me | 3.43 | 2.009 | 0.837 | 0.693 |
| The use of artificial intelligence enhances my tourist experience | 3.35 | 2.002 | 0.833 | 0.793 |
| Using artificial intelligence has become a habit for me | 3.43 | 2.073 | 0.826 | 0.864 |
| I like the AI application | 3.68 | 2.070 | 0.844 | 0.694 |

**Table 2.** *Cont.*

| Items | m | sd | $\alpha$ | Factor Loading |
|---|---|---|---|---|
| AI application is kind of fun for me | 3.43 | 2.009 | 0.837 | 0.693 |
| The use of artificial intelligence enhances my tourist experience | 3.35 | 2.002 | 0.833 | 0.793 |
| Using artificial intelligence has become a habit for me | 3.43 | 2.073 | 0.826 | 0.864 |
| I have to use AI when I travel | 1.99 | 0.164 | 0.852 | 0.782 |
| I am addicted to using AI for its sustainability benefits | 1.57 | 0.503 | 0.868 | 0.931 |
| Using artificial intelligence has become commonplace for me | 2.00 | 0.151 | 0.819 | 0.815 |
| I intend to continue using AI in the future to contribute to sustainable business | 2.00 | 0.303 | 0.856 | 0.637 |
| I plan to continue to use AI frequently to improve my work | 1.96 | 1.217 | 0.888 | 0.941 |
| I foresee the use of artificial intelligence in the near future for the benefit of sustainable business | 3.46 | 1.212 | 0.831 | 0.723 |
| I want to inform others to use artificial intelligence when they travel | 2.58 | 1.440 | 0.877 | 0.658 |
| I want to continuously improve AI technology | 3.62 | 1.381 | 0.826 | 0.801 |
| I very often use artificial intelligence to plan work in a hotel | 2.02 | 1.322 | 0.819 | 0.970 |
| I very often use artificial intelligence to plan tourism products | 2.02 | 1.204 | 0.810 | 0.623 |

Note: m—arithmetic mean, sd—standard deviation, $\alpha$—Cronbach's alpha.

Table 3 presents the results of the factor analysis conducted on the measured constructs. Each row corresponds to a specific construct, including performance expectancy, effort expectancy, social influence, facilitating conditions, hedonic motivation, habit, behavioral intention, and AI usage behavior. The table provides the mean (m) and standard deviation (sd) of the responses, the Cronbach's alpha ($\alpha$) coefficient for internal consistency reliability, the percentage of variance explained by each factor, the cumulative percentage of variance explained, the composite reliability (CR), and the average variance extracted (AVE). The results indicate the satisfactory internal consistency of the measurements, as confirmed by the high values of Cronbach's alpha coefficient ($\alpha$) for each construct. Additionally, the identified factors explain a significant percentage of variance in responses (ranging from 24.514% to 66.306%), indicating an adequate measurement model.

**Table 3.** The measurement items, descriptive statistics, and reliability analysis of the constructs for the factors.

| Factors | m | sd | $\alpha$ | % of Variance | Cumulative % | CR | AVE |
|---|---|---|---|---|---|---|---|
| Performance expectancy | 2.41 | 0.895 | 0.655 | 24.514 | 24.514 | 0.800 | 0.505 |
| Effort expectancy | 2.24 | 1.00 | 0.677 | 10.598 | 35.112 | 0.810 | 0.520 |
| Social influence | 3.43 | 1.072 | 0.663 | 8.181 | 43.293 | 0.838 | 0.564 |
| Facilitating conditions | 3.29 | 1.591 | 0.632 | 6.983 | 50.276 | 0.813 | 0.523 |
| Hedonic motivation | 3.47 | 1.699 | 0.645 | 4.410 | 54.686 | 0.824 | 0.541 |
| Habit | 1.89 | 0.196 | 0.734 | 4.095 | 58.780 | 0.911 | 0.722 |
| Behavioral intention | 2.09 | 0.834 | 0.740 | 3.999 | 62.780 | 0.858 | 0.561 |
| AI usage behavior | 1.98 | 0.968 | 0.746 | 3.562 | 66.306 | 0.847 | 0.656 |

Note: m—arithmetic mean, sd—standard deviation, $\alpha$—Cronbach's alpha, CR—composite reliability, AVE—average variance extracted, % of variance—the percentage of variance explained by the extracted factors, cumulative %—cumulative share of variance explained by the extracted factors, AI usage behavior describes the actual behavior of users in connection with the use of artificial intelligence in the direction of sustainability, behavioral intention describes the user's intention or willingness to use AI in the future with the idea of potential sustainability.

## 4. Results

Table 4 gives an insight into the reliability and validity of the construct. The R squared ($R^2$) table provides insights into the explained variance and the impact of individual factors on the dependent variables. The $R^2$ values indicate the proportion of variance explained by the model for each dependent variable [109]. In the case of behavioral intention, the $R^2$ is 0.683, suggesting that approximately 6.83% of the variance in behavioral intention is explained by the model. The adjusted $R^2$, which considers the number of predictors in the

model, is 0498. For AI usage behavior, the $R^2$ is higher, at 0.456, indicating that about 45.6% of the variance in AI usage behavior is explained by the model. The adjusted $R^2$ is 0.455.

**Table 4.** The construct's reliability and validity.

| Factors | Cronbach's Alpha (>0.6) | rho_A (>0.7) | CR (>0.7) | AVE (>0.5) |
|---|---|---|---|---|
| Behavioural intention | 0.691 | 0.771 | 0.808 | 0.660 |
| Facilitating conditions | 0.864 | 0.841 | 0.816 | 0.537 |
| Hedonic motivation | 0.786 | 0.726 | 0.919 | 0.733 |
| Performance expectancy | 0.700 | 0.706 | 0.863 | 0.609 |
| AI Usage behavior | 0.729 | 0.738 | 0.847 | 0.651 |
| Effort expectancy | 0.742 | 0.715 | 0.832 | 0.553 |
| Habit | 0.715 | 0.733 | 0.887 | 0.677 |
| Social influence | 0.674 | 0.882 | 0.914 | 0.727 |
| Behavioral Intention | | | AI Usage Behavior | |
| $R^2$ | $R^2$ adjusted | | $R^2$ | $R^2$ adjusted |
| 0.683 | 0.498 | | 0.456 | 0.455 |

Note: CR—composite reliability; AVE—average variance extracted; rho_A—omega reliability.

The results show that all constructs in the research have satisfactory reliability and validity. Examining various measures of reliability and validity, it is observed that all constructs exceeded the critical thresholds for Cronbach's alpha (all values above 0.6), rho_A (all values above 0.7), composite reliability (all values above 0.7) and the average variance extracted (AVE) (all values over 0.5). This indicates that each construct, behavioral intention, facilitating conditions, hedonic motivation, performance expectancy, AI usage behavior, effort expectancy, habit, and social influence, has good internal consistency to reliably measure the intended concepts. High values of composite reliability and the AVE also confirm that the constructs are well defined and that the relevant indicators are effective in measuring those constructs. These results provide a solid basis for further analysis and the interpretation of data in the context of this research.

The results of the combined Fornell–Larcker criterion and HTMT analysis offer strong evidence of discriminant validity across all constructs, as the square roots of the AVE values for each construct exceed the HTMT values in comparisons with all other constructs. This outcome firmly establishes that each construct in the model is significantly distinct and does not share excessive variance with other constructs, underscoring their unique contributions to the research model. In Table 5, the diagonal shaded in gray visually emphasizes the square roots of AVE values, crucial for the Fornell-Larcker criterion, confirming that each construct uniquely contributes without excessive shared variance with others, contrasting with the HTMT measure of similarity among constructs.

**Table 5.** The check of discriminant validity using Fornell–Larcker and HTMT criteria.

| | Behavioral Intention | Facilitating Conditions | Hedonic Motivation | Performance Expectancy | AI Usage Behavior | Effort Expectancy | Habit | Social Influence |
|---|---|---|---|---|---|---|---|---|
| Behavioral intention | 0.812 | 0.061 | 0.087 | 0.106 | 0.727 | 0.148 | 0.199 | 0.104 |
| Facilitating conditions | 0.061 | 0.733 | 0.120 | 0.400 | 0.035 | 0.539 | 0.075 | 0.309 |
| Hedonic motivation | 0.087 | 0.120 | 0.856 | 0.556 | 0.066 | 0.498 | 0.095 | 0.370 |
| Performance expectancy | 0.106 | 0.400 | 0.556 | 0.780 | 0.063 | 0.132 | 0.117 | 0.414 |
| AI usage behavior | 0.727 | 0.035 | 0.066 | 0.063 | 0.807 | 0.086 | 0.271 | 0.082 |
| Effort expectancy | 0.148 | 0.539 | 0.498 | 0.132 | 0.086 | 0.744 | 0.113 | 0.283 |
| Habit | 0.199 | 0.075 | 0.095 | 0.117 | 0.271 | 0.113 | 0.823 | 0.112 |
| Social influence | 0.104 | 0.309 | 0.370 | 0.414 | 0.082 | 0.283 | 0.112 | 0.853 |

The variance inflation factor (VIF) values assess multicollinearity among the variables in the model. These values indicate the extent to which the variance of an estimated regression coefficient increases when the predictors are correlated (Table 6).

**Table 6.** Collinearity statistics (variance inflation factor—VIF).

| Factors | Items | Variance Inflation Factor—VIF (VIF < 3.3) |
|---|---|---|
| AI usage behavior | AIUB1 | 1.660 |
| | AIUB2 | 1.917 |
| | AIUB3 | 1.285 |
| Behavioral intention | BI1 | 1.071 |
| | BI2 | 1.656 |
| | BI3 | 1.228 |
| | BI4 | 1.842 |
| Effort expectancy | EE1 | 1.478 |
| | EE2 | 1.732 |
| | EE3 | 1.531 |
| | EE4 | 1.231 |
| Facilitating conditions | FC1 | 2.280 |
| | FC2 | 2.575 |
| | FC3 | 2.197 |
| | FC4 | 1.666 |
| Habit | HBT1 | 2.337 |
| | HBT2 | 1.034 |
| | HBT3 | 3.314 |
| | HBT4 | 1.869 |
| Hedonic motivation | HM1 | 1.482 |
| | HM2 | 1.393 |
| | HM3 | 2.142 |
| | HM4 | 1.711 |
| Performance expectancy | PE1 | 1.200 |
| | PE2 | 2.174 |
| | PE3 | 1.338 |
| | PE4 | 1.991 |
| Social influence | S1 | 1.209 |
| | S2 | 1.157 |
| | S3 | 2.430 |
| | S4 | 2.570 |

All VIF values are below the recommended threshold of 3.3, indicating adequate independence between variables. Based on this, we conclude that the measurement of the structural model is reliable and valid in terms of the multicollinearity test.

Based on the above values, it can be concluded that the saturated and estimated models are extremely similar in terms of fit. All values (SRMR, d_ULS, d_G, Chi-Square, and NFI) are identical or very close between the two models, indicating that the estimated model provides almost the same level of fit as the saturated model (Table 7). This suggests that the estimated model effectively follows the structure of the data. A high NFI and low values for the SRMR, d_ULS, and d_G indicate good model fit. The Chi-Square values are low, which also indicates a good fit [109].

**Table 7.** Fit summary indices.

|  | Saturated Model | Estimated Model |
|---|---|---|
| SRMR | 0.072 | 0.072 |
| d_ULS | 0.069 | 0.069 |
| d_G | 0.041 | 0.043 |
| Chi-Square | 2.782 | 2.782 |
| NFI | 0.961 | 0.961 |

Figure 2 displays the direct influences between variables in a model that explores the factors affecting the intention and actual usage behavior of artificial intelligence. Lower value connections are also included, allowing for a more comprehensive insight into all influences within the model. Indirect influences are not shown in this figure but are presented in Table 8.

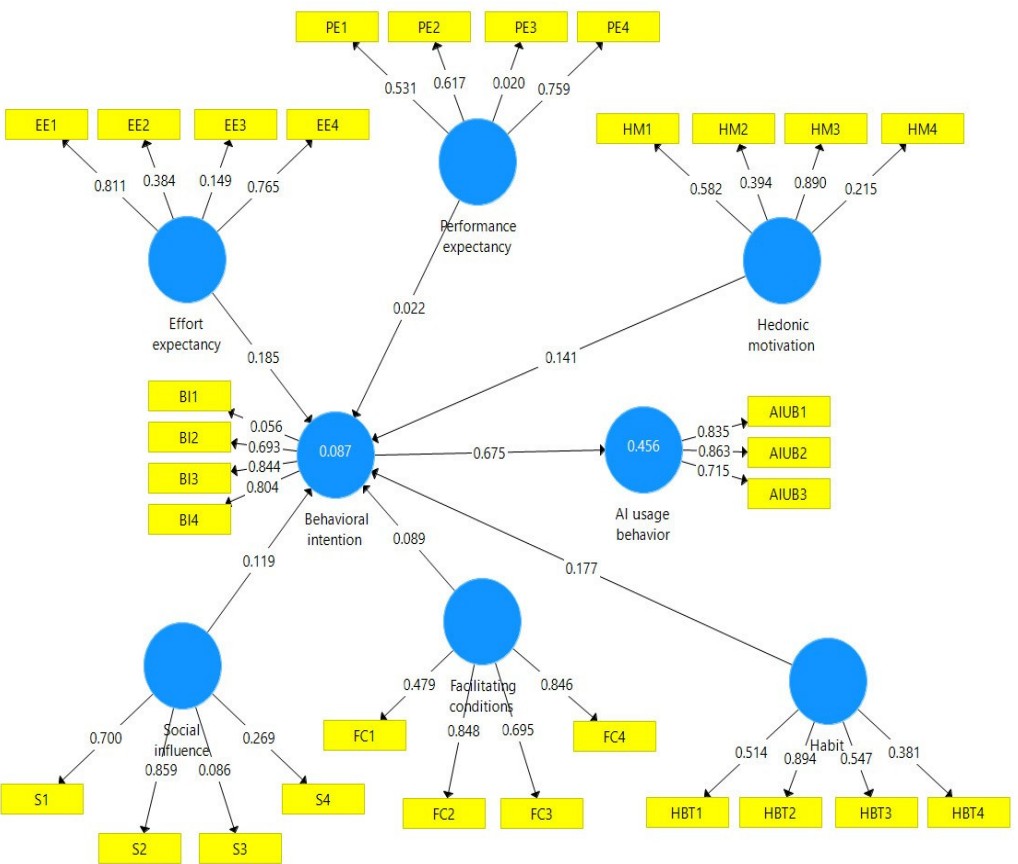

**Figure 2.** Path coefficient estimates.

Table 8 shows the estimated path coefficients (β) between the various factors, the sample means (means), and the sample standard deviations for each factor. Also, the table contains information on the t-values and *p*-values used to test the statistical significance of the relationships between factors. In particular, the indirect effect, as well as the influence of behavioral intention as a mediator, is highlighted. The same table provides a detailed overview of the analysis of the hypotheses set in the research.

Behavioral intention ➜ AI usage behavior (β = 0.675; *p* = 0.000; H1 confirmed).

**Table 8.** Values of the estimates, means, standard deviations, t-statistics, and *p*-values of the structural model.

| | Estimate (β) | Sample Mean (M) | Standard Deviation | Path from BI to AUB (β) | Indirect Effect (β) | t Statistics | *p* Values | Hypothesis |
|---|---|---|---|---|---|---|---|---|
| Behavioral intention ➜ AI Usage behavior | 0.675 | 0.678 | 0.016 | - | - | 42.543 | 0.000 | H1 ✔ |
| Facilitating conditions ➜ Behavioral intention | 0.089 | 0.000 | 0.089 | 0.675 | | 0.995 | 0.020 | H2 ✔ |
| Hedonic motivation ➜ Behavioral intention | 0.141 | 0.011 | 0.120 | 0.675 | 0.09517 | 1.176 | 0.040 | H3 ✔ |
| Performance expectancy ➜ Behavioral intention | 0.022 | 0.013 | 0.072 | 0.675 | 0.01485 | 0.312 | 0.055 | H4 ✔ |
| Effort expectancy ➜ Behavioral intention | 0.185 | 0.126 | 0.154 | 0.675 | 0.01248 | 1.196 | 0.032 | H5 ✔ |
| Habit ➜ Behavioral intention | 0.177 | 0.187 | 0.038 | 0.675 | 0.01194 | 4.712 | 0.000 | H6 ✔ |
| Social influence ➜ Behavioral intention | 0.119 | 0.030 | 0.109 | 0.675 | 0.06007 | 1.096 | 0.054 | H7 ✔ |

Note: ✔—hypothesis confirmed.

Based on this result, it can be inferred that there is a significant mediating role between the behavioral intentions and the actual usage of artificial intelligence (AI) among hotel employees in Serbia. The high beta coefficient (β = 0.675) suggests that as employees' intentions to use AI increase, their actual usage of AI also significantly increases. The *p*-value (<0.001) confirms the statistical significance of this relationship, affirming H1. This finding underscores the critical role of cultivating strong behavioral intentions towards AI usage for its effective adoption in the hotel industry. The mediation analysis revealed the nuanced ways in which facilitating conditions, hedonic motivation, effort expectancy, habit, performance expectancy, and social influence impact the usage of AI through behavioral intention. Facilitating conditions, despite their modest effect (β = 0.089), still play a crucial role in nurturing the intention to use AI, thereby underscoring the importance of providing the right support and resources for fostering AI engagement. Hedonic motivation emerges as a more potent indirect influencer (β = 0.141), signifying that the pleasure and positive experiences associated with AI significantly bolster the willingness to use it, subsequently translating into actual usage behavior. Effort expectancy is highlighted as a key driver (β = 0.185), suggesting that the perceived ease of using AI shapes the intention to engage with it and is a critical consideration for enhancing AI adoption. Furthermore, the habitual use of AI (β = 0.177) indicates that as employees become more accustomed to incorporating AI into their daily routines, their continued usage is likely. This points to habit formation as a strategic element in ensuring sustained AI application in the workplace. Meanwhile, performance expectancy and social influence, although positive, exhibit weaker effects on behavioral intention, signaling that while these factors contribute to the shaping of intentions towards AI usage, they may not be as compelling as the other predictors. Overall, these findings offer valuable insights into the complex interplay of the factors influencing AI's adoption and highlight where managerial interventions could be most effective.

Facilitating conditions ➜ Behavioral intention (β = 0.089; *p* = 0.020; H2 confirmed)

The positive beta coefficient (β = 0.089) shows a statistically significant but relatively modest influence of facilitating conditions on behavioral intention. While facilitating conditions do impact employees' intentions to use AI, the strength of this effect is less pronounced. The significance (*p* = 0.020) validates the relationship, suggesting that while important, facilitating conditions alone may not be the strongest predictor of the behavioral intention to use AI. Enhancing facilitating conditions could still contribute to more favorable attitudes towards AI adoption.

Hedonic motivation ➜ Behavioral intention (β = 0.141; *p* = 0.040; H3 confirmed).

With a beta coefficient of 0.141 and a *p*-value of 0.040, hedonic motivation has a significant positive effect on behavioral intention. This indicates that the more employees

find AI enjoyable and satisfying to use, the stronger their intention is to adopt it. This effect, while statistically significant, highlights the importance of the user experience in promoting AI adoption in the hotel industry.

Performance expectancy ➜ Behavioral intention (β = 0.022; *p* = 0.055; H4 marginally confirmed).

The relationship between performance expectancy and behavioral intention is positive but with a very low beta coefficient (β = 0.022), and the *p*-value (0.055) is slightly above the traditional threshold for significance. This result suggests a very weak and marginally non-significant influence of performance expectancy on behavioral intention, indicating that expectations regarding the performance of AI might not be a strong driver of the behavioral intention to use AI among hotel employees. This relationship, being borderline significant, warrants further investigation.

Effort expectancy ➜ Behavioral intention (β = 0.185; *p* = 0.032; H5 confirmed).

Effort expectancy has a positive and significant effect on behavioral intention, with a beta coefficient of 0.185. This means that the easier the AI technology is to use, the stronger the employees' intentions are to use it. The significant *p*-value (0.032) confirms this relationship, highlighting the importance of user-friendly AI technologies in fostering adoption intentions.

Habit ➜ Behavioral intention (β = 0.177; *p* = 0.000; H6 confirmed).

Habit shows a significant positive impact on behavioral intention (β = 0.177), indicating that the more ingrained the use of AI becomes in the employees' daily routines, the stronger their intention is to use AI. The highly significant *p*-value (<0.001) underscores the critical role of habit formation in the adoption of AI technologies.

Social influence ➜ Behavioral intention (β = 0.119; *p* = 0.054; H7 marginally confirmed).

With a beta coefficient of 0.119, social influence appears to have a positive effect on the behavioral intention to use AI, although this effect is only marginally non-significant (*p* = 0.054). This suggests that the opinions and behaviors of others may somewhat influence employees' intentions to use AI, but this influence is not as strong or as clear-cut as some other factors. This borderline significance indicates a potential area for further exploration.

In presenting our SEM findings, we note the inclusion of items with lower loading values. This decision was informed by a balanced evaluation of theoretical significance and empirical evidence, underscoring our commitment to a nuanced exploration of AI adoption dynamics. Despite their lower loadings, these items contribute to a richer, more textured understanding of the factors at play, offering valuable insights into the diverse attitudes and behaviors towards AI within the hotel industry.

A bootstrapping analysis was conducted to examine the identified relationships (structural model) (Figure 3). This involved randomly selecting observations from the original data set (with replacement) to create subsamples. These subsamples were then used to calculate a PLS path model. This procedure was repeated multiple times to generate a significant number of random subsamples.

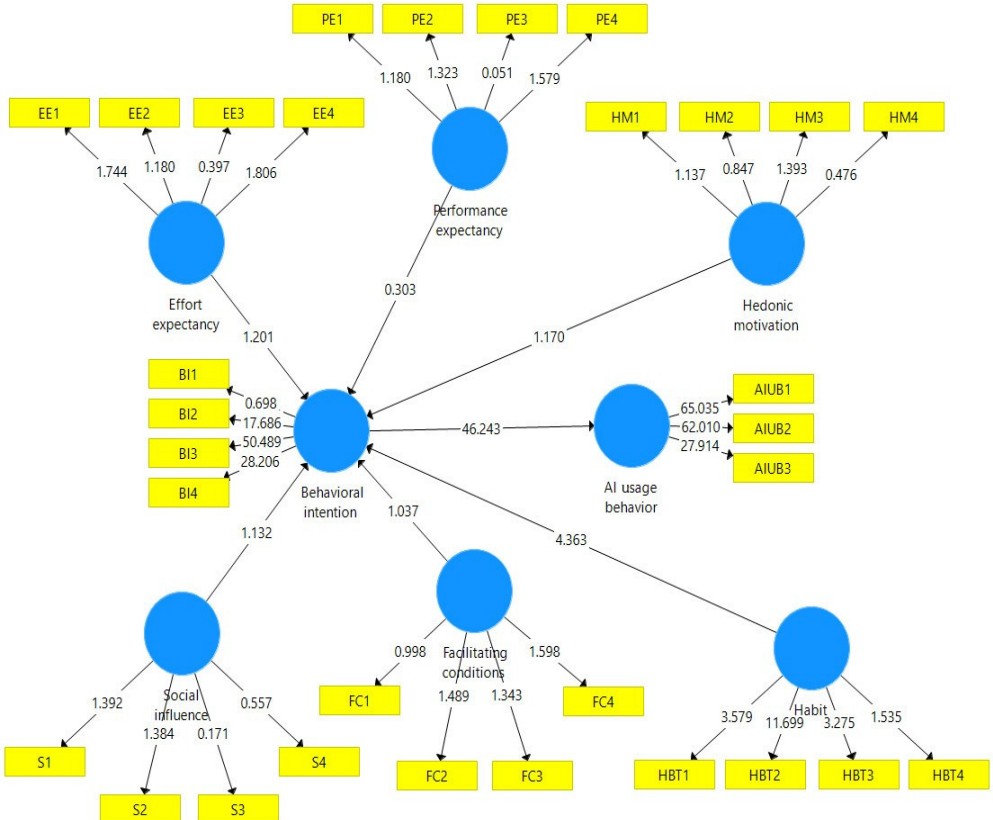

**Figure 3.** The path model with the bootstrapping results.

In the model, each construct is measured across a set of indicators, and the coefficients between indicators and constructs indicate how strongly each indicator represents the construct it measures. The coefficients are high, which indicates that the indicators well "capture" what the construct should represent, which further means that the measurement is reliable. The bootstrapping method helped us confirm that these coefficients are robust, that is, they can be reliably used in further analyses and conclusions.

## 5. Discussion

In this research, we analyze how hotel employees in Serbia accept the application of artificial intelligence (AI), a key technology in the modern hotel industry. This research aims not only to understand current trends, but also to assess how the application of AI can contribute to the potential sustainability of hotel operations. The seven hypotheses we put forward shed light on various aspects and factors that influence this process.

In the analysis of the structural model, H1 reveals a pronounced effect ($\beta = 0.675$) in the mediation relationship between various predictive factors (facilitating conditions, effort expectancy, hedonic motivation, social influence, and habit) and AI usage behavior, with behavioral intention acting as the intermediary. The mediation analysis of our study highlights the pivotal role of behavioral intention in bridging the gap between various predictors and the actual usage of AI in the hotel industry. The indirect effects observed from facilitating conditions, hedonic motivation, effort expectancy, and habit confirm that while the direct path to AI usage is important, their relation through behavioral intention is crucial.

Unlike the dominant influence of H1, other hypotheses that examine various predictors of behavioral intention show considerably weaker effects. The effect estimates for these hypotheses fall within a relatively lower range, illustrating that although each of these factors contributes to the formation of behavioral intention, their individual contributions are far less compared to the direct impact of this intention on AI usage. This difference in

effect strengths underscores the importance of understanding how behavioral intention acts as a key mediator between various psychological and situational factors and actual AI usage behavior.

Furthermore, facilitating conditions in the workplace exhibit a statistically significant but relatively modest influence on employees' behavioral intentions towards AI adoption ($\beta$ = 0.089, *p* = 0.020), in line with H2. While favorable conditions contribute to shaping intentions, their impact may be overshadowed by other determinants. Our results also highlight the significance of hedonic motivation in fostering behavioral intention towards AI adoption ($\beta$ = 0.141, *p* = 0.040), supporting H3. Employees who derive pleasure and satisfaction from using AI technologies are more likely to intend to adopt them, emphasizing the experiential aspect of AI acceptance.

However, the influence of performance expectancy on behavioral intention appears to be marginal and non-significant ($\beta$ = 0.022, *p* = 0.055), providing limited support for H4. This suggests that employees' expectations regarding AI performance may not strongly drive their intentions to adopt it. Conversely, factors such as effort expectancy ($\beta$ = 0.185, *p* = 0.032) and habit ($\beta$ = 0.177, *p* < 0.001) demonstrate significant positive effects on behavioral intention, supporting H5 and H6, respectively. User-friendly interfaces and established usage habits play crucial roles in fostering employees' intentions towards AI adoption. While social influence exhibits a positive but marginally significant effect on behavioral intention ($\beta$ = 0.119, *p* = 0.054), partially supporting H7, its influence appears to be less pronounced compared to other factors.

Our decision to retain certain items with lower statistical loadings in the SEM analysis warrants further discussion. This choice was not made lightly but was rooted in a firm belief in the items' theoretical contributions to their respective constructs. By retaining these items, we aimed to encapsulate the complexity of the constructs fully, thereby enriching our analysis and discussion of AI adoption in the hospitality sector. We acknowledge the potential methodological concerns this decision may raise; however, we assert that the inclusion of these items adds depth to our study, facilitating a more comprehensive exploration of AI's role in advancing sustainability and efficiency within the Serbian hotel industry.

This exploration into AI adoption within the Serbian hotel industry has yielded valuable insights, providing answers to the research questions posed in the introductory section. The results obtained indicate that employees in the Serbian hotel industry generally hold positive perceptions towards the acceptance and use of AI technologies. They perceive AI to be a valuable tool that can enhance efficiency, improve service quality, and contribute to innovation within the industry. These perceptions are influenced by various factors such as its ease of use, perceived benefits, and the organizational support for AI integration.

The results reveal that employees' perceptions towards AI significantly influence their actual usage behavior. When employees perceive AI technologies positively, with high levels of intention and motivation to use them, they are more likely to actively engage with these technologies in their daily work practices. Conversely, negative perceptions or skepticism towards AI may result in lower levels of adoption and usage among employees.

The findings highlight the critical role of facilitating conditions, such as access to resources and support from management, in shaping employees' perceptions and behaviors towards AI adoption. Effort expectancy, reflecting the perceived ease of use of AI technologies, influences employees' willingness to adopt and engage with these technologies. Similarly, performance expectancy, which relates to the perceived benefits and effectiveness of AI, also impacts employees' perceptions and behaviors towards AI adoption. Social influence, including peer opinions and organizational norms, can further shape employees' attitudes and intentions towards AI. Additionally, hedonic motivation, or the enjoyment derived from using AI, can positively influence employees' perceptions and behaviors towards AI adoption, fostering greater acceptance and usage.

The responses suggest that the adoption of AI technologies in the Serbian hotel industry has the potential to significantly contribute to sustainability goals. By streamlining

operations, optimizing resource utilization, and enhancing service delivery, AI can help hotels minimize waste, reduce energy consumption, and mitigate their environmental impact [112–116]. Moreover, AI-driven analytics and predictive modeling can facilitate data-driven decision-making, leading to more efficient resource allocation and improved environmental stewardship. Overall, the adoption of AI technologies has the potential to align with the sustainability objectives of the Serbian hotel industry, fostering economic, social, and environmental benefits.

## 6. Conclusions

The application of artificial intelligence (AI) in hotel businesses opens up new opportunities for improving sustainability in this industry. This approach not only contributes to increasing efficiency and reducing costs, but also plays an important role in improving the quality of service and satisfaction of both employees and guests [117–119]. Based on the analyzed results, we can conclude that there is a significant relationship between various factors and the use of artificial intelligence (AI). First and foremost, we observe a strong correlation between behavioral intention and actual AI usage behavior. This suggests that when users have a clear intention to use AI, they are more likely to act accordingly. Facilitating factors also play a role, but to a lesser extent, in shaping behavioral intention and actual AI use behavior. This indicates that providing more favorable conditions may have some influence on encouraging users to use AI, but it is not a decisive factor. Hedonic motivation, although present, does not show a significant association with behavioral intention. This suggests that the enjoyment and satisfaction users get from using AI does not play a key role in their decision to use it. Performance expectancy and effort expectancy are relatively less important in the formation of behavioral intention. This means that users may not consider how effective or easy-to-use AI is when deciding to use it. Habit, as a factor, has a significant impact on behavioral intention and actual AI usage behavior. This highlights the importance of creating a routine in the use of AI as a means of increasing its use. Finally, social influence has a moderate effect on behavioral intention, indicating that other people's opinions and attitudes may play a certain, but not decisive role in the decision to use AI.

In the realm of the results questioning the impact of facilitating conditions and expected performance on the application of artificial intelligence (AI) in the hotel industry, the need for more detailed research becomes apparent. It is especially important to understand the specifics of these factors within the context of the hotel industry in the Republic of Serbia. The complexity of employee attitudes and behaviors underscores that it is not enough to simply introduce AI technologies; it is also necessary to deeply understand how these factors operate at the local level. For the successful integration of AI into the hotel industry of Serbia, a holistic approach that respects technological, human, organizational and market aspects is necessary. Such an approach can enable not only technological adaptation, but also the creation of a sustainable and innovative business model that will respond to the challenges and take advantage of the opportunities brought by the digital era. Certainly, the results of this research, which talk about the acceptance of the application of AI in Serbian hotels, can in some ways represent the basis for creating a theory about sustainable hotel business.

### 6.1. Theoretical Implications

This study directs attention to the importance of behavioral intention as a key factor in the adoption of artificial intelligence (AI) in the Serbian hotel industry. Emphasizing the applicability of behavior theories, the study advocates for different interventions in the attitudes and intentions of employees. The investigation into the variable impact of facilitating conditions on employees' attitudes provides a starting point for the reassessment of theoretical frameworks and the exploration of additional factors influencing attitude formation and intentions. The focus on the balanced relationship between habits and behavioral intentions, along with the insignificant link between habits and the usage of AI,

reveals a complex dynamic that necessitates further theoretical exploration. Understanding the interaction between habit formation in shaping intentions and subsequent behaviors allows for a more detailed investigation into the dynamics of artificial intelligence adoption. Exploring the negligible connection between expected performance and behavioral intentions underscores the need for more advanced explorations of the factors shaping perceptions of the utility of artificial intelligence, potentially uncovering hidden dynamics in employee attitudes. Additionally, the results of this study provide a foundation for the development of a missing theory regarding the assumption that the greater use of AI affects the sustainability of hotel businesses in Serbia. This theory would deal with examining how the increased use of AI in the hospitality industry can contribute to environmental, economic, and social sustainability. This includes analyzing how AI can aid in optimizing operations, reducing environmental impacts, improving guest and employee satisfaction, and enhancing market competitiveness. Although certain results pertaining to the influence of some factors on AI adoption exhibit relatively weak effects, they have been retained within the model for a deliberate purpose. Retaining these variables allows for a more comprehensive exploration within the literature, facilitating comparative analyses and theoretical extensions across diverse contexts. By retaining even marginally significant findings, our study contributes to the broader theoretical discourse surrounding AI adoption dynamics, fostering a different understanding of the multifaceted factors influencing organizational behaviors and attitudes towards emerging technologies. This study, therefore, highlights the need for a multidisciplinary approach that integrates knowledge from technology, management, sociology, and ecology fields to understand the complex relationships between the use of AI and sustainability in the hotel industry.

### 6.2. Practical Implications

For managers in the Serbian hotel industry, these findings offer significant insights that can be strategically applied to advance the adoption of artificial intelligence. Encouraging positive behavioral intentions through targeted interventions can significantly influence how employees interact with and accept AI technology. To do this effectively, organizations need to develop tailored strategies that go beyond simply providing enabling conditions. This requires a deeper understanding of employee perceptions, attitudes, and the unique challenges they face in their roles. Recognizing the role of habit formation in shaping intentions implies that organizations could greatly benefit from initiatives that actively cultivate positive habits around the use of AI. This can be achieved through various means such as comprehensive training programs, engaging in awareness campaigns, and designing AI user interfaces that are intuitive and easy to adopt. These initiatives should be ongoing, not one-off events, to reinforce positive habits and ensure long-term behavioral change. Moreover, the insignificant impact of expected performance on AI adoption highlights the need for organizations to expand their focus beyond traditional performance metrics. When promoting AI adoption, it is important to communicate and highlight the broader benefits of AI technologies. Employee training and education programs should be structured to not only impart their technical knowledge, but also address their potential misconceptions and foster a more comprehensive understanding of the role of artificial intelligence in improving overall operational efficiency, guest experiences, and hotel sustainability.

Fostering an organizational culture that values innovation and adaptability is key. This includes leadership support for AI initiatives, creating a safe environment for experimentation and feedback, and recognizing and rewarding employees who contribute to the successful integration of AI into their workflows. Given the dynamic and competitive nature of the hotel industry, the continuous monitoring and evaluation of the effectiveness of AI solutions and strategies is essential. This entails collecting employee feedback, assessing customer satisfaction, and monitoring technological advances to ensure AI solutions remain relevant and effective.

It should be emphasized that keeping factors within the model, even those that show marginal effects on AI usage behavior, offers significant practical benefits for organizations. Firstly, maintaining a comprehensive model allows organizations to understand the myriad of factors influencing employees' attitudes towards AI technologies. While certain factors may seem insignificant individually, their collective impact within organizational dynamics is crucial. By retaining these factors, organizations can identify subtle potential interactions that may influence AI integration success. Factors that appear insignificant in isolation could exert indirect or interactive effects when considered together. Furthermore, retaining non-significant factors fosters a robust approach to organizational learning and adaptation. Acknowledging diverse influences encourages a culture of openness to change, promoting continuous improvement and innovation in AI adoption practices.

### 6.3. Limitations

Although this study provides valuable insights into the willingness of employees in the Serbian hotel industry to embrace artificial intelligence, it is important to recognize certain limitations and contextual factors. Primarily, this research concentrates on the hotel industry within Serbia, and its findings may not be immediately applicable to other sectors or different cultural contexts. The cross-sectional nature of the study limits the ability to establish causal relationships, and a longitudinal approach could yield a more nuanced understanding of these dynamics over time. Another limitation is the reliance on self-reported measures, which may introduce response bias. Future research could benefit from incorporating a combination of qualitative and quantitative methodologies to enhance the robustness and depth of the findings.

Moreover, while the study effectively illustrates the readiness of employees to accept AI in hotel operations, it does not directly investigate its impact on sustainability. Generally, proving sustainability in business operations can be challenging, and the study primarily aimed to demonstrate employee readiness for AI adoption rather than its direct influence on sustainable practices. This is an important distinction, as the readiness to adopt AI does not inherently guarantee sustainability outcomes. Additionally, the study did not explore potential external factors, such as economic conditions, regulatory influences, or technological advancements, which may significantly impact the adoption and effective utilization of AI in the Serbian hotel industry. These external factors can play a crucial role in shaping both the opportunities and challenges associated with AI adoption.

Given these considerations, while the study lays a foundation for understanding AI adoption in the Serbian hotel industry, it also highlights the need for further research. This research should aim to bridge these gaps, particularly focusing on the long-term sustainability impacts of AI integration and the influence of external environmental factors. Such investigations could provide more comprehensive insights, aiding in the formulation of strategies that not only foster AI adoption but also promote sustainable development within the industry.

We are aware of the methodological lack of our study, notably our approach to item retention in the face of lower loading values. This decision underscores our dedication to a thorough and theoretically grounded investigation of AI adoption. We believe that this approach not only strengthens the integrity of our findings but also lays a foundation for future research to build upon, encouraging further inquiry into the subtleties of technology adoption in hospitality.

### 6.4. Future Directions and Global Implications

Future research endeavors could explore the temporal dynamics of artificial intelligence's adoption into the hotel industry in Serbia, tracking changes in attitudes and behaviors over an extended period. Comparative studies across different industries and countries could offer a broader understanding of contextual variations in AI's adoption. Investigating the role of external factors and contextual influences in shaping attitudes towards artificial intelligence might uncover hidden determinants not addressed in this

study. Moreover, the integration of qualitative methodologies, such as interviews and focus groups, could provide richer insights into the nuanced experiences and perceptions of employees in the Serbian hospitality sector. This study lays the groundwork for a more nuanced understanding of artificial intelligence's adoption into the hotel industry in Serbia, offering theoretical insights and practical implications. Recognizing limitations and identifying future directions will contribute to the ongoing discourse on the role of artificial intelligence in organizational environments. Given the complexity of employees' attitudes, we conclude that the successful integration of artificial intelligence into the hotel industry requires a comprehensive approach that takes into account individual differences, organizational support, and market dynamics—all while being mindful of the overarching goals of sustainability. These findings provide a foundation for further research and the development of AI adoption strategies in the specific context of the hotel sector in Serbia. It emphasizes the continuous pursuit of sustainable development within the industry.

To further enhance the strategic approach towards sustainable AI integration, the incorporation of these findings into a broader sustainability strategy for the hotel industry in Serbia is essential. This would involve creating a roadmap that aligns AI adoption with sustainable business practices, focusing on areas such as energy efficiency, waste reduction, and improved customer experiences. The strategy should consider both short-term and long-term goals, identifying key performance indicators that can measure the impact of AI on various aspects of sustainability. Collaboration with stakeholders, including government bodies, industry associations, and technology providers, will be crucial in this endeavor. Their involvement can facilitate the sharing of best practices, access to resources, and the creation of a supportive ecosystem for AI integration.

In expanding the relevance of our findings beyond the Serbian hotel industry, it is important to consider the generalizability of our results. While our study provides nuanced insights into the adoption and implications of AI within Serbian hotels, these insights may also resonate with broader, international contexts. Factors influencing AI adoption, such as performance expectancy and effort expectancy, are likely to be relevant in other countries, especially those with similar economic and technological landscapes.

To contextualize these factors internationally, future research could compare the determinants of AI adoption in Serbian hotels with those of hotels in other countries, particularly within the Central and Eastern European region where the economic and cultural factors may be comparable. Furthermore, a cross-cultural study examining the differences and similarities in AI's adoption across various hospitality markets could provide a more comprehensive understanding of the global implications of our findings.

As the global hotel industry increasingly embraces technology in its operations and customer service, the insights gained from our study about employee attitudes and usage behaviors could inform international strategies for AI's implementation. This would be particularly relevant for multinational hotel chains seeking to standardize AI's integration across different countries while being mindful of local nuances.

The potential for AI to contribute to sustainable business practices, as highlighted by our findings, also warrants international consideration. With sustainability being a global concern, understanding how AI can drive such practices in the hotel industry has implications for policy-making and strategic planning on an international scale.

**Author Contributions:** Conceptualization, T.G. and D.V.; methodology, F.Đ.; software, J.B.; validation, A.S. and S.K.; formal analysis, J.Đ.B.; investigation, T.G.; resources, D.V.; data curation, S.M.; writing—original draft preparation, T.G. and S.G.; writing—review and editing, L.D.D.; visualization, J.Đ.B.; supervision, A.S. All authors have read and agreed to the published version of the manuscript.

**Funding:** This research received no external funding.

**Institutional Review Board Statement:** Not applicable.

**Informed Consent Statement:** Not applicable.

**Data Availability Statement:** Data are contained within the article.

**Conflicts of Interest:** The authors declare no conflicts of interest.

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
