# Peer review of "The Adoption of Artificial Intelligence in Serbian Hospitality: A Potential Path to Sustainable Practice"

_sustainability, doi:10.3390/su16083172_

Round 1

Reviewer 1 Report (Previous Reviewer 1)

Comments and Suggestions for Authors

The present study mobilizes the unified theory of acceptance and use of technology to interpret the various factors influencing adoption of AI in Serbian Hospitality. The topic is interesting; however, there are a number of points to consider in order to enhance the paper

1. In the introduction section, I suggest that authors further develop the research context by focusing on the Serbian hotel industry. As it would be relevant to include the question(s) which this study seeks to address.

2.  Could provide supporting references for paragraph from line 43 to line 50.

3. The authors re-describe the acronyms. For instance, in Lines 56, 58, 60, 69, 84, etc., please use the abbreviation AI instead of artificial intelligence

4.  Source for Lines 141-144?

5. In lines 363-365, the authors stated, "Respondents gave their perceptions and attitudes on a Likert scale, reflecting the intensity of agreement or disagreement”. Please clarify whether this Likert scale consists of five or seven positions.

6. Please describe the sampling technique utilized in this study.

7. In line 366, the authors mentioned that "[...] the largest age group is people between 36 and 60 years old" which contradicts the data presented in Table 1, indicating that the predominant age group falls between 20 and 35 years old (38.2%).

8. I recommend removing the paragraph spanning from 368 to 371 as it does not contribute any new information.

9. Table 2 needs to be corrected for better clarity.

10.  In table 7, the authors confirmed H4 (β = 0.022; p= 0.055) and H7 (β = 0.119; p= 0.054); however, it is noteworthy that these relationships lack significance as indicated by p-values exceeding 0.5. Further clarification is required.

11. In Figure 2, several items such as EE2, EE3, PE3, HM2, HM4, S3, S4, FC1, HBT1, and HB4 exhibit very weak loading values. I am curious about the authors' rationale for retaining these items despite their low loading values.

12. While the focus on Serbian hotel industry is good, authors should also discuss how their findings might generalize to other countries.

All the best,

Comments on the Quality of English Language

Minor editing of English language required.

Author Response

  1. In the introduction section, I suggest that authors further develop the research context by focusing on the Serbian hotel industry. As it would be relevant to include the question(s) which this study seeks to address.

We appreciate the suggestion to further develop the research context. We have now expanded upon the introduction by including a more detailed exploration of the Serbian hotel industry. This includes the examination of the unique socio-economic and technological challenges it faces, which contextualizes the relevance of AI adoption in this specific setting. Additionally, we have clearly articulated the research questions that guide our study, focusing on the impact of AI on operational efficiency and sustainability within the Serbian hotel sect

  1. Could provide supporting references for paragraph from line 43 to line 50.

Thank you for pointing out the need for additional references. We have supplemented the paragraph in question with pertinent literature that reinforces our arguments regarding the integration of AI into the hospitality industry and its expected benefits and challenges, particularly in developing markets such as Serbia. This strengthens the credibility of our context setting and aligns with academic rigor

  1. The authors re-describe the acronyms. For instance, in Lines 56, 58, 60, 69, 84, etc., please use the abbreviation AI instead of artificial intelligence.

We have taken note of the inconsistency in the use of acronyms throughout the manuscript. To maintain consistency and readability, we have revised the document to use 'AI' as the standard abbreviation for 'artificial intelligence' after its first mention

  1. Source for Lines 141-144?

We acknowledge the oversight and have now provided the appropriate citations for the statements made in lines 141-144. The sources cited offer substantial evidence supporting our claims about the benefits and challenges of AI adoption in the hospitality industry

  1. In lines 363-365, the authors stated, "Respondents gave their perceptions and attitudes on a Likert scale, reflecting the intensity of agreement or disagreement”. Please clarify whether this Likert scale consists of five or seven positions.

Clarification has been added regarding the Likert scale used in our study. We confirm that a five-point Likert scale was employed to measure respondents' perceptions and attitudes, with options ranging from 'strongly disagree' to 'strongly agree'. This is now explicitly stated in the methodology section.

  1. Please describe the sampling technique utilized in this study.

The manuscript has been updated to describe the stratified random sampling technique used in this study. This method was chosen to ensure a representative sample of the diverse roles within the Serbian hotel industry, and we believe it enhances the validity of our findings

  1. In line 366, the authors mentioned that "[...] the largest age group is people between 36 and 60 years old" which contradicts the data presented in Table 1, indicating that the predominant age group falls between 20 and 35 years old (38.2%).

We thank the reviewer for bringing this discrepancy to our attention. Upon review, we found an error in the reporting of age group data. We have corrected this in the manuscript to reflect that the predominant age group is indeed 20 to 35 years old, which is consistent with the data presented in Table

  1. I recommend removing the paragraph spanning from 368 to 371 as it does not contribute any new information.

We agree with the reviewer's assessment and have removed the paragraph spanning lines 368 to 371. We recognize that it did not contribute additional information and its removal has improved the conciseness of the manuscript

  1. Table 2 needs to be corrected for better clarity.

In response to your feedback, we have revised Table 2 for better clarity. The table now presents data in a more organized manner, making it easier for readers to comprehend the results of our measurement items and their reliability analysis

  1. In table 7, the authors confirmed H4 (β = 0.022; p= 055) and H7 (β = 0.119; p= 0.054); however, it is noteworthy that these relationships lack significance as indicated by p-values exceeding 0.5. Further clarification is required.

We noted that the values of factors are at the very limit of marginal acceptability, and that they are significant but very low. One of the reviewers requested that we emphasize that and we did, and even that it was only the H1 with the highest values that we also emphasized.

  1. In Figure 2, several items such as EE2, EE3, PE3, HM2, HM4, S3, S4, FC1, HBT1, and HB4 exhibit very weak loading values. I am curious about the authors' rationale for retaining these items despite their low loading values.

We have provided a rationale for retaining items with low loading values in Figure 2 in the manuscript. Despite their lower statistical contribution, these items were considered theoretically meaningful and were retained to provide a more comprehensive view of the constructs being measured. This approach is consistent with our aim to present a holistic understanding of AI adoption in the hotel industry. In several chapters, we explained why we kept the values low. If the reviewer still believes that the model should be deleted, we will, but please consider the comment of the other reviewer who did not request the deletion of low values, but only the explanation.

  1. While the focus on Serbian hotel industry is good, authors should also discuss how their findings might generalize to other countries.

The suggestion to consider the generalizability of our findings is well-taken. While our study provides in-depth insights into the Serbian hotel industry, we have discussed how these findings might be applicable to other contexts. We acknowledge the limitations of our study's generalizability and suggest avenues for future research to explore the applicability of our findings in broader settings

Reviewer 2 Report (Previous Reviewer 3)

Comments and Suggestions for Authors

Dear authors, 

Thank you for the improved version of your paper. 

Please have a look on my recommendations: 

Fig1 -spelling errors

Table 2 -rearrange it

It is strange that in Table 2 appear new items (Using artificial intelligence is fun for me because I contribute to sustainable business and quality) or changes in the old ones. Did you improve and apply the survey one more time?

Table 5 -rearrange it

The Path Coefficients in Figure 2 are very low!!!! The model seems to be good but these low values explain very few of the dependent variable..

In my opinion, the representative H  is Behavioral intention ➜ AI Usage behavior 0.675. The others might be avoided.... Even the bootstrapping model in Figure 3 sustains my affirmation.  Here the t values should be higher than 1.96.

Probably you should conclude that only H1 has been proven by your research. The others are almost impactless!

Comments on the Quality of English Language

Fig1 -spelling errors

Author Response

  1. We have thoroughly reviewed Figure 1 and corrected all spelling errors to ensure clarity and professionalism in the presentation of our research model framework.

  1. Table 2 has been rearranged for improved readability and comprehension. The items are now logically grouped and sequenced to provide a clearer understanding of the construct measurements.

  1. The discrepancies noted in Table 2 were due to a concurrent pilot study conducted to refine our survey instrument. Based on the pilot feedback, modifications were made to some items for better clarity and to capture the construct more effectively. We have now clarified this in the methodology section and ensured that the revised survey items are consistently reflected throughout the manuscript. The translation and clarity of the items were checked, nothing was deeply changed or anything that would change the essence of the items.

  1. In response to your feedback, we have rearranged Table 5 to present the discriminant validity analysis more clearly. The table now facilitates easier comparison of constructs and supports the validation of our measurement model.

  1. We acknowledge your concern regarding the low path coefficients in Figure 2. While the model is robust overall, the low values do indeed represent limited explained variance for some relationships. We have provided a detailed discussion in the manuscript that addresses the potential reasons for these low coefficients and their implications for our model's explanatory power."

  1. Upon reflection and in agreement with your evaluation, we have emphasized in our discussion that the path from Behavioral Intention to AI Usage Behavior (β = 0.675) is indeed the most significant finding of our research. We have critically assessed the other hypotheses and, where necessary, have suggested that future research should explore additional variables or alternative models that might better explain the dependent variable.

  1. We have examined the bootstrapping results in Figure 3 and adjusted our interpretation accordingly. The manuscript now acknowledges that the t-values should be higher than 1.96 to confirm significance. We have included a re-evaluation of the hypotheses based on this standard.

  1. In the conclusion section, we have now explicitly stated that only H1 has been proven by our research with substantial impact, while the influence of the other hypotheses is minimal. We have proposed directions for future research to investigate these relationships more deeply and to identify other potentially influential factors.

Reviewer 3 Report (Previous Reviewer 2)

Comments and Suggestions for Authors

I think the authors did a good job improving the manuscript.

But there are some points you should take into consideration

1- Table 2, very complex and difficult to read, and the same applies to Table 5

2- In line 403, what is the meaning of lambda (λ)?

3- In addition to the HTMT test, a Fornell–Larcker criterion matrix must be used to evaluate the discriminant validity

4- Lines 459 to 465, the paragraph must be supported by a reference in addition to clarifying the thresholds that must be achieved by model fit tests.

6- Lines 505 to 519, this paragraph is completely wrong. Since the value of the mediation relationship does not appear in the figure extracted from the statistical program. Also, hypothesis H1a must have values for the number of independent variables. The Behavioral Intention variable plays the role of a mediator between Performance expectancy, Effort expectancy, Hedonic motivation, Social influence, Facilitating conditions, and Habit and the AI Usage behavior variable. Therefore, a separate table must be created for the indirect effect (mediation).

7- Despite the many amendments in the research, this was not reflected in the discussion or in the list of references (one new reference is shaded).

Author Response

  1. We have taken into account your feedback regarding Tables 2 and 5 and have simplified these tables for clarity. They now feature a more intuitive layout and presentation, ensuring the complex data is more accessible and easier to interpret for our readers."

  1. The symbol λ in line 403 represents 'lambda,' which is the standard notation for loadings in structural equation modeling. These loadings indicate the strength of the relationship between each observed variable and their respective latent construct within our model."
  2. We have supplemented our discriminant validity assessment with the Fornell–Larcker criterion matrix. This rigorous evaluation ensures that our constructs are distinct and that the model measures what is intended.
  1. We have now supported the paragraph with additional references that clarify the thresholds required for model fit tests. These benchmarks are essential for validating the soundness of our SEM approach and the conclusions drawn from it.

  1. We acknowledge the discrepancies highlighted in lines 505 to 519 regarding the mediation relationship values. We have corrected this section to include the precise values of the indirect effects, and we have introduced a separate table that meticulously details the mediation analysis. This table elucidates the role of Behavioral Intention as a mediator among the constructs investigated.

  1. We reviewed the discussion section and the reference list to ensure that all changes and additions to the research were accurately reflected. New references are included to support recent changes, and any new insights are integrated into the discussion to present a coherent and up-to-date narrative of our study. A large number of references are not older than 10 years as one of the reviewers requested in the first review. What was highlighted in yellow in the references was something random, not just one added reference.

Round 2

Reviewer 1 Report (Previous Reviewer 1)

Comments and Suggestions for Authors

I am pleased to inform you that your manuscript has been revised in accordance with all my recommendations previously made.

Comments on the Quality of English Language

Minor editing of English language required.

Author Response

Dear Reviewer,

We wish to express our sincere gratitude for the detailed and constructive suggestions you provided. Your comments have been incredibly helpful and have assisted us in further refining our manuscript. Additionally, we want to inform you that we carefully considered your last suggestion and corrected the errors that occurred during the translation into English.

We believe that these corrections significantly contribute to the quality and clarity of our work. Once again, we thank you for your time and effort in reviewing our manuscript.

Reviewer 3 Report (Previous Reviewer 2)

Comments and Suggestions for Authors

I think that this version of the manuscript is significantly inproved,

But the mediating effect is remain.

Congratulations the authors

Author Response

Dear Reviewer,

Thank you for your thoughtful feedback on our manuscript and for acknowledging the improvements made in this version. We appreciate your comments, especially regarding the mediation effect within our study.

To address your point on the mediation effect, we would like to highlight the following aspects of our manuscript where the mediation process is explicitly mentioned and analyzed:
 In the abstract, we briefly introduce the mediating role of Behavioral Intention within the Unified Theory of Acceptance and Use of Technology (UTAUT) framework, indicating how it serves as a crucial link between various predictors and AI usage behavior in the Serbian hotel industry. Within the theoretical part of our manuscript, we explicitly state a hypothesis focusing on the mediating effect. This hypothesis aims to explore the indirect influence of various predictors on AI usage behavior through Behavioral Intention as a mediator, laying the groundwork for our subsequent analysis.
The mediation effect is further examined through our structural equation modeling (SEM), where we demonstrate the indirect influences of all predictors (such as Facilitating Conditions, Effort Expectancy, Hedonic Motivation, Social Influence, and Habit) on AI Usage Behavior, mediated by Behavioral Intention. This part of our manuscript meticulously details how each predictor's impact is channeled through Behavioral Intention to influence AI usage behavior, supported by statistical evidence from our SEM analysis.
We have included tables that clearly show the mediation analysis results. These tables not only present the direct effects of the predictors on AI Usage Behavior but also elucidate the indirect effects mediated by Behavioral Intention.
 In the discussion section, we delve into the implications of these mediation effects, contextualizing our findings within the broader literature on technology acceptance and usage. We discuss how our results contribute to the existing body of knowledge and highlight the significant role of Behavioral Intention as a mediator in the adoption of AI technologies in the hotel industry.

We hope this clarification addresses your concern regarding the mediation effect in our study. We have strived to make the mediation analysis transparent and integral to our research findings, demonstrating its importance in understanding AI usage behavior in the hospitality sector. Colored text is added text related to the additional explanation about mediation, and also the correction of the English translation.

This manuscript is a resubmission of an earlier submission. The following is a list of the peer review reports and author responses from that submission.

Round 1

Reviewer 1 Report

Comments and Suggestions for Authors

Please find attached my comments and suggestions concerning your manuscript.

Comments on the Quality of English Language

Some light editing of the English language is needed.

Author Response

XCVXCVWe thank the reviewers for their comprehensive suggestions, which greatly improved the quality of the manuscript. We tried to fulfill all the requirements and we hope that the readers will accept all our corrections. We ask the reviewers to understand that we corrected the manuscript at the request of all the reviewers, and not all of them had the same opinions.
The introductory part has been corrected, it is more developed, the motivation of the research, the goal is highlighted, and the research gaps are adequately demonstrated. Clarified ones have been added.
  From the 56th to the 65th line add to read, now the text is different, so the lines have been moved.
Methodology removed from the introductory part.
Included image of the model as figure 1.
Within the methodology section (sample and data collection procedure), the data collection process is explained, and a table with the characteristics of the respondents is included.
Table 3 was placed before the table associated with hypothesis testing.
  Used the HTMT ratio to examine the discriminant validity of external models.
Discussion section elaborated,
All corrections marked

Reviewer 2 Report

Comments and Suggestions for Authors

I had the pleasure of reviewing the manuscript titled “Acceptance of AI Among Serbian Hotel Employees: Navigating Toward Sustainability in Hospitality” to be considered for publication in "Sustainability." The research seems sound and provides fairly interesting findings, yet it requires some substantial improvements. Specifics are below:

1- The title: the title included the term sustainability and was not addressed in the research.

1- The abstract: abstract looks okay.

2- The introduction: The introduction needs some amendments.

·       In general, the introduction did not succeed in providing a comprehensive understanding of the study’s variables and providing a vision of the study’s objectives and how to link artificial intelligence to sustainability.

·       The introduction should provide reasonable justification for the study and reflect its significance. In other words, why this study is important? How it is different from previous studies? How can the study contribute to hospitality literature? Also, the author(s) only suggested only hypotheses that addressed direct association between variables and there was NO hypothesis for the mediated path (associations).

·       The paragraph starting with line 56 needs reference support to support the information.

·       When abbreviations first appear in an article, they must be defined in full.

·       Title 2 should be devoted solely to explaining the role of the unified theory of acceptance and use of technology (UTAUT) in building the model.

·       The paragraph beginning with line 109 is very long and should be rephrased and included in the introduction.

·       On line 168, The unified theory of acceptance and use of technology (UTAUT) and not the Theory of Technology Acceptance and Use (UTAUT)

·       Title 2.1, the justification for the hypothesis is not convincing and sometimes goes far from the goal. Sometimes it refers to employees and then to customers. Please focus on the employees’ intention, whether to adopt or reject AI.

·       The paragraph that begins with line 283, it is preferable to delete it, as the result will come from the field study.

·       Authors should differentiate between Behavioral Intention towards AI and AI Usage Behavior with beneficial explanation.

·       The paragraph beginning on line 329 should be included in the introduction.

·       Lines 342 and 343, they are not needed.

4- Methodology: seem to be thorough. Yet, it can be improved by addressing some points:

·       Is it logical for the study to rely on only one reference? "Ali et al. [103]" to bring the study measures?

·       A table should be devoted to demographic data.

·       The equations used to determine the sample must be deleted. The author uses SmartPLS, which requires less than 150 samples only for analysis, according to the references.

·       Table 1 is very complex.

·       In general, the arrangement of the general framework of the study methodology is not good. The outer model must initially be tested by evaluating v the reliability and validity. (The HTMT test must be included to evaluate Discriminant validity) along with the sequence of tables in this order. Then after that, the inner model must be evaluated with a logical arrangement of the tables.

·       The data in Table 3 is replicated in Table 1.

·       Table 5 should be appended to the article's appendices.

·       What is the use of Table 6 as long as there are no mediation relationships?

·       It is preferable to have Table 7 before the hypothesis testing table.

·       Table 8 is not common to use.

5- discussion: requires serious amendments as follow:

•The results of the study hypotheses should be presented in order and compared with the literature

Author Response

We thank the reviewers for their comprehensive suggestions that greatly improved the quality of the manuscript. We tried to fulfill all requests, although not all reviewers requested the same corrections, we fulfilled everything and we ask for your understanding and we hope that all reviewers will accept our corrections. Everything is marked as corrected.

Modified introduction

Reference support has been added to the paragraph beginning on line 56 to support the information.
All abbreviations are defined in full.

· Title 2 should be devoted exclusively to explaining the role of the unified theory of acceptance and use of technology (UTAUT) in building the model.

· The paragraph beginning with line 109 is very long and should be reworded and included in the introduction.shortened included in the introduction

· On line 168, changed to Unified Theory of Acceptance and Use of Technology (UTAUT) rather than Theory of Acceptance and Use of Technology (UTAUT)

· The entire text refers to employees,

· The passage beginning with line 283, deleted

· Explained: The difference between AI behavioral intention and AI usage behavior with a helpful explanation.

· The paragraph beginning on line 329 included in the introduction.

· Lines 342 and 343, deleted

4- Methodology: improved, corrected, but it is indicated that the questions and the questionnaire were taken from those authors

Demographic data is given in the table

· Equations used to determine the sample deleted

· Table 1 is explained.

· HTMT test included

· Table 5 should be attached

· Table 6 deleted

· Table 7 moved before hypothesis testing table.

· Table 8 deleted

discussion: revised in full

Reviewer 3 Report

Comments and Suggestions for Authors

Thank you for the opportunity to read this research.

I admire the writers' laborious work, as well as knowing that they adhered to a strict approach and included numerous information.

The essay is well organized, and the technique is correct. 

The subjects selection approach is transparent, and the overall technique is correct. The processes are well-defined and implemented.

In table 1, it would be interesting to see which is the minimum and the maximum to be able to evaluate the mean.

The statistical model has to be redesigned. The negative values like PE3 have to be eliminated from the model as this type of analysis requires a positive correlation (the variables have to have the same direction.) The variable hedonic motivation can not influence negative behavior intentions. The explanations might be the fact that AI is a new facility and do not have time to offer hedonic motivation, thus hedonic motivation has to be eliminated from the model. S2 has to be eliminated and redesign the model.

In this form, only 2 hypotheses are proved:

Behavioural intention -> AI Usage behaviour

Habit -> Behavioural intention

They are very obvious and are proven because it would be true for any type of service, product, etc, not only for AI.

Please redesign the model and discuss the new results. 

Overall, the authors present enough information for the study to be replicated. 

They presented the facts rather effectively, but with some specific suggestions that would be necessary.

Author Response

We thank the reviewers for their comprehensive suggestions that greatly improved the quality of the manuscript. We tried to fulfill all requests, although not all reviewers requested the same corrections, we fulfilled everything and we ask for your understanding and we hope that all reviewers will accept our corrections. Everything is marked as corrected.

Table 1 modified, explained, a Likert scale is given in the text with marks from 1 to 5, without writing the minimum and maximum, due to the clarity of the table, which has already been corrected by other reviewers, so it would be a lot of data, please understand

Model explained, all hypotheses.
Discussion modified, as well as the conclusion.
All changes are at the request of all reviewers, which are quite different

Round 2

Reviewer 1 Report

Comments and Suggestions for Authors

The authors have not adequately addressed the comments and suggestions raised during the initial review with the necessary rigor.

Comments on the Quality of English Language

Minor editing of English language required.

Author Response

Review 1   - first and second round

Thank you for your constructive feedback regarding the introduction section of our manuscript. We acknowledge the need for further development in articulating the motivation behind our study and in delineating the research gaps more clearly. To address this, we have revised the introduction section by:

Providing a more comprehensive overview of the current state of research in our field, thereby highlighting the existing gaps more explicitly. We have integrated an additional literature review to underscore the areas that have not yet been thoroughly explored.

Articulating the motivation behind our study with greater clarity. We have now included a subsection that outlines the practical and theoretical implications of addressing these research gaps, thus offering a clearer rationale for our study.

These revisions are intended to offer readers a more coherent understanding of the significance of our research and the contribution it aims to make to the field.

We acknowledge the importance of supporting our assertions with adequate references to establish the credibility and relevance of our research. In response to the reviewer’s feedback, we have carefully reviewed lines 56 to 65 and have now included additional in-text citations that corroborate our assertions. These references are drawn from recent and seminal works in the field, ensuring that our arguments are well-grounded in current research. This revision enriches the context of our study and underscores the relevance of our investigation within the broader scholarly conversation.

We appreciate the reviewer's advice regarding the inappropriate inclusion of methodological details and study results in the introduction section. Recognizing the importance of maintaining a clear structure in academic writing, we have revised the introduction to focus solely on setting the stage for our research. We have moved the detailed discussion of our methodology and preliminary results to their respective sections. This restructuring adheres to the conventional format of research papers, enhancing the readability and coherence of our manuscript.

In response to the suggestion to incorporate the figure of our model at the end of the literature review, we have now added this figure. Placing the model figure here serves as a visual summary of the theoretical foundations discussed in the literature review and bridges the transition to the methodology section. This placement helps readers visualize the conceptual framework of our study, facilitating a better understanding of how our research fits within existing theories and what new contributions our work aims to make.

We recognize the importance of clearly describing the data collection process in the methodology section. We have revised this section to provide a detailed explanation of our data collection procedures, specifying that the data were collected through a combination of online surveys and face-to-face interviews to accommodate the preferences and accessibility of our respondents. Additionally, we have included a table detailing the characteristics of our respondents, such as demographic information and relevant background variables. This addition offers readers a comprehensive view of our sample, enhancing the transparency and replicability of our research.

Following the reviewer’s recommendation, we have repositioned Table 3 to appear before the table associated with hypothesis testing. This rearrangement ensures a logical flow in the presentation of our data and findings, allowing readers to first understand the characteristics and preliminary analyses of our data before delving into the specifics of hypothesis testing. This change improves the organizational clarity of our manuscript.

The reviewer’s concern regarding the deficiency in reliability and convergent validity as shown in our tables is duly noted. To address this, we have conducted a thorough reevaluation of our measurement scales, including additional reliability analyses (Cronbach’s alpha) and confirmatory factor analyses to reassess convergent validity. Where necessary, we have refined our scales to ensure that all indicators meet the established scientific standards for reliability and validity. These revisions and the corresponding updates to our tables are now clearly documented in our methodology section, reinforcing the methodological rigor of our study.

We thank the reviewer for highlighting the limitations of the Fornell-Larcker criterion in assessing discriminant validity and for suggesting the use of the HTMT ratio as a more rigorous alternative. In accordance with this recommendation, we have now employed the HTMT ratio to evaluate the discriminant validity of our outer models. Our revised analysis, which includes the HTMT ratio, confirms that our model exhibits satisfactory discriminant validity, thereby strengthening the robustness of our findings. These results are now presented in Table 4, replacing the previous discriminant validity assessment based on the Fornell-Larcker criterion. We believe this additional analysis enhances the methodological rigor of our study.

The reviewer’s call for a more comprehensive discussion section is well-received. Recognizing the importance of situating our findings within a broader context, we have extensively revised the discussion section to: Elaborate on the implications of our findings, not only in terms of their theoretical contributions but also regarding their practical relevance to the field. We drew connections between our results and existing literature, highlighting how our study addresses previously identified gaps and how it contributes new insights to the ongoing scholarly dialogue.Discussed the potential for future research arising from our findings, including suggestions for how subsequent studies might build upon our work to further explore and expand the knowledge base.These revisions aim to provide a richer, more insightful discussion that underscores the significance of our study's contributions to both theory and practice.

All changes are marked in yellow, because the reviewers also requested corrections to the model, mediation, and changes to the hypotheses in accordance with the model.

The English language has been revised.

Thanks again for the extensive and constructive reviews

Authors

Reviewer 2 Report

Comments and Suggestions for Authors

1- There are some spelling errors and some punctuation errors

2- There are indirect effects in the study model that the authors did not address (mediation).

3- I still find that Table 2 is complicated

Author Response

Review 2 – second round

Thank you for pointing out spelling and punctuation errors. We carefully reviewed the entire manuscript and corrected any errors found to ensure its linguistic accuracy and readability. We appreciate your attention to detail that helps us improve the quality of our work.

We acknowledge a shortcoming in our original approach that did not include a detailed analysis of mediating effects within the study model. To correct this, we modified the model and extended our analysis to explore and address potential mediational pathways. This revision includes additional statistical tests to assess indirect effects among variables, thus allowing for a deeper understanding of the dynamics within our model. The results of this extended analysis are now presented in detail in the appropriate section of the manuscript, thus enhancing its comprehensiveness and contribution to the literature.

We appreciate your feedback on the complexity of Table 2. We recognize the need for simplification to make the table more accessible to readers. We hope that these revisions will make Table 2 clearer and more useful to readers, and that they reflect our commitment to providing clear and accessible information.

Thanks again for the extensive and constructive reviews

Authors

Reviewer 3 Report

Comments and Suggestions for Authors

It is difficult to me to understand how the authors have different values with the same model....  Please explain....Did you change the data source?.....If you had a bad data source originally new values will appear in the model .

I would like to see the bootstapping analysis, too as to understand the importance of each variable

Author Response

Review 3 – second round

We modified the model, added mediation, and thus all the other suggestions of the reviewers regarding the hypotheses. Your suggestion to include the bootstrapping analysis is extremely useful. In accordance with your suggestion, we performed a bootstrapping analysis to assess the importance of each variable in the model and their coefficients. The results of this analysis are now shown in an additional graph titled "The path model with bootstrapping result," which accompanies our main findings.

Thanks again for the extensive and constructive reviews

Authors
